# Propylene Production via Oxidative Dehydrogenation of Propane with Carbon Dioxide over Composite M_x_O_y_-TiO_2_ Catalysts

**DOI:** 10.3390/nano14010086

**Published:** 2023-12-28

**Authors:** Alexandra Florou, Georgios Bampos, Panagiota D. Natsi, Aliki Kokka, Paraskevi Panagiotopoulou

**Affiliations:** 1Laboratory of Environmental Catalysis, School of Chemical and Environmental Engineering, Technical University of Crete, GR-73100 Chania, Greece; aflorou@tuc.gr (A.F.); akokka@tuc.gr (A.K.); 2Department of Chemical Engineering, University of Patras, GR-26504 Patras, Greece; geoba@chemeng.upatras.gr (G.B.); natsi@chemeng.upatras.gr (P.D.N.)

**Keywords:** oxidative dehydrogenation of propane with CO_2_, composite metal oxides, TiO_2_, Ga, Cr, Zr, Ce, Ca, surface basicity

## Abstract

The CO_2_-assisted oxidative dehydrogenation of propane (ODP) was investigated over titania based composite metal oxides, 10% M_x_O_y_-TiO_2_ (M: Zr, Ce, Ca, Cr, Ga). It was found that the surface basicity of composite metal oxides was significantly higher than that of bare TiO_2_ and varied in a manner which depended strongly on the nature of the M_x_O_y_ modifier. The addition of metal oxides on the TiO_2_ surface resulted in a significant improvement of catalytic performance induced by a synergetic interaction between M_x_O_y_ and TiO_2_ support. Propane conversion and propylene yield were strongly influenced by the nature of the metal oxide additive and were found to be superior for the Cr_2_O_3_-TiO_2_ and Ga_2_O_3_-TiO_2_ catalysts characterized by moderate basicity. The reducibility of the latter catalysts was significantly increased, contributing to the improved catalytic performance. This was also the case for the surface acidity of Ga_2_O_3_-TiO_2_ which was found to be higher compared with Cr_2_O_3_-TiO_2_ and TiO_2_. A general trend was observed whereby catalytic performance increased significantly with decreasing the primary crystallite size of TiO_2_. DRIFTS studies conducted under reaction conditions showed that the adsorption/activation of CO_2_ was favored on the surface of composite metal oxides. This may be induced by the improved surface basicity observed with the M_x_O_y_ addition on the TiO_2_ surface. The Ga_2_O_3_ containing sample exhibited sufficient stability for about 30 h on stream, indicating that it is suitable for the production of propylene through ODP with CO_2_ reaction.

## 1. Introduction

Propylene is a versatile precursor for the formation of various derivatives used in our daily life (e.g., polypropylene, isopropanol, acrylic acid, acrylonitrile, propylene oxide, butyraldehyde, cumene, etc.) and, thus, it is considered as a key component of the chemical industry [1,2]. One of the traditional methods used for propylene production is the reaction of propane dehydrogenation (1), which is strongly endothermic and equilibrium limited [3]. The reaction requires high temperatures, and, therefore, high energy consumption, suffering from fast catalyst deactivation as well as low C_3_H_8_ conversions and C_3_H_6_ selectivities [4]. Oxidative dehydrogenation of propane in the presence of molecular oxygen has been proposed as an alternative pathway. This is an exothermic reaction, with no thermodynamic limitations, and operable at low reaction temperatures. The main drawback of this process is the deep oxidation of both C_3_H_8_ and C_3_H_6_ towards CO and CO_2_, resulting in low propylene yields [4]. Thus, the replacement of molecular oxygen by a milder and readily available oxidant, such as CO_2_, has recently gained interest as an alternative approach for selective propylene production [1,3,4,5,6,7]. This approach has the advantage that CO_2_ participates both in (a) propane conversion towards propylene (2) and (b) hydrogen consumption via the reverse water–gas shift (RWGS) reaction (3) [8]. Removing hydrogen from the gas stream can overcome the equilibrium limitations of propane dehydrogenation, resulting in higher propylene yields [1,6,9]. Moreover, carbon monoxide produced via both reactions is a valuable byproduct, which can be utilized in chemical synthesis [5,6].
(1)$$C_{3}H_{8}\;↔\;C_{3}H_{6}+H_{2}\;{\Delta H}_{298Κ}^{0}\;=124.3\;\text{kJ/mol}$$
(2)$${\mathrm{CO}}_{2}+C_{3}H_{8}\;↔\;C_{3}H_{6}+\mathrm{CO}+H_{2}O\;{\Delta H}_{298Κ}^{0}\;=165.4\;\text{kJ/mol}$$(3)$${\mathrm{CO}}_{2}+H_{2}\;↔\;\mathrm{CO}+H_{2}O\;{\Delta H}_{298Κ}^{0}\;=41.1\;\text{kJ/mol}$$

Depending on the catalyst and reaction conditions employed, the reactions of propane hydrogenolysis (4) and (5), and propane or propylene decomposition (6)–(9), may also take place, resulting in a decrease in propylene yield and possibly surface carbon formation (8) and (9) [3,4,7]. Dry reforming of propane may also run in parallel leading to syngas production (CO/H_2_) (10) [10,11].
(4)$$C_{3}H_{8}+H_{2}\;↔\;C_{2}H_{6}+{\mathrm{CH}}_{4}\;{\Delta H}_{298Κ}^{0}\;=-55.4\;\text{kJ/mol}$$
(5)$$C_{3}H_{8}+2H_{2}\;↔\;3{\mathrm{CH}}_{4}\;{\Delta H}_{298Κ}^{0}\;=-120.0\;\text{kJ/mol}$$
(6)$$2{\mathrm{CO}}_{2}+2C_{3}H_{8}\;↔\;3C_{2}H_{4}+2\mathrm{CO}+2H_{2}O\;{\Delta H}_{298Κ}^{0}\;=447.2\;\text{kJ/mol}$$
(7)$$C_{3}H_{8}\;↔\;C_{2}H_{4}+{\mathrm{CH}}_{4}\;{\Delta H}_{298Κ}^{0}\;=81.7\;\text{kJ/mol}$$
(8)$$2C_{3}H_{6}\;↔\;2{\mathrm{CH}}_{4}+C_{2}H_{4}+2C_{\left(s\right)}\;{\Delta H}_{298Κ}^{0}\;=-137.6\;\text{kJ/mol}$$
(9)$$C_{3}H_{8}\;↔\;{\mathrm{CH}}_{4}+2H_{2}+2C_{\left(s\right)}\;{\Delta H}_{298Κ}^{0}\;=29.2\;\text{kJ/mol}$$
(10)$${\mathrm{CO}}_{2}+3C_{3}H_{8}\;↔\;6\mathrm{CO}+4H_{2}\;{\Delta H}_{298Κ}^{0}\;=644.1\;\text{kJ/mol}$$

Carbon dioxide may also be involved in the reverse Boudouard reaction (11), removing coke from the catalyst surface and, thus, improving catalyst stability [5].
(11)$${\mathrm{CO}}_{2}+C\;↔\;2\mathrm{CO}\;{\Delta H}_{298Κ}^{0}\;=172.4\;\text{kJ/mol}$$

The major benefit of the ODP process is the utilization of CO_2_, of which emissions into the atmosphere have increased rapidly during recent decades and nowadays is considered as one of the main greenhouse gases resulting in global warming and, therefore, major climate change [4,12]. However, CO_2_ is a thermodynamically stable compound (ΔG_f_ = −394 kJ∙mol^−1^), the reduction of which requires high energy reactants combined with active and selective catalysts as well as optimal reaction conditions to gain a thermodynamic driving force. Thus, in order for the proposed process to be effective, (a) a suitable catalyst must be applied to selectively promote both the ODP with CO_2_ and RWGS reactions and be able to retard C_3_H_8_ and/or C_3_H_6_ decomposition and hydrogenolysis reactions, and (b) operating conditions should be optimized.

Oxidative dehydrogenation of propane with CO_2_ has been investigated over various single or composite metal oxides, including MnO [13], Cr_2_O_3_/SiO_2_ [14,15], Cr_2_O_3_/ZrO_2_ [16], Cr_2_O_3_/Al_2_O_3_ [14,16], V_2_O_5_/SiO_2_ [17], Ga_2_O_3_ [18], Ga_2_O_3_/TiO_2_ [15], Ga_2_O_3_/Al_2_O_3_ [15,19], Ga_2_O_3_/ZrO_2_ [15,20], Ga_2_O_3_/SiO_2_ [15], Ga_2_O_3_/MgO [15], noble metal catalysts supported on metal oxides (e.g., Pt/Al_2_O_3_ [7], Au/ZnO [3], Pd/CeZrAlO_x_ [21]), as well as zeolites with different frameworks [4]. The beneficial effect of CO_2_ on catalytic performance varies depending on the catalyst employed. For example, in the case of Ga_2_O_3_ based catalysts, CO_2_ (a) suppresses catalyst deactivation by carbon deposition due to the occurrence of the reverse Boudouard reaction, (b) enhances propylene yield by removing H_2_ via the RWGS reaction [6,18], and (c) favors the desorption of propylene from the catalyst surface [22]. The beneficial effect of CO_2_ over Cr_2_O_3_ based catalysts has been related to CO_2_ involvement in subsequent reduction–oxidation cycles between Cr^6+^ and Cr^3+^, which have been found to be crucial in the propane dehydrogenation pathway [6,23]. In the case of ceria based catalysts, CO_2_ has the dual role of regenerating selective oxygen species, and shifting the equilibrium for propane dehydrogenation by consuming H_2_ through the RWGS [21]. In particular, the lattice oxygen ions abstract hydrogen from propane molecules to form propylene and H_2_O, while CO_2_ replenishes these selective oxygen species, releasing CO in the gas phase. Clearly, the design and development of new catalytic materials for the ODP reaction requires a detailed investigation of CO_2_ interaction with catalytic active sites in order to elucidate the exact role of CO_2_ on the reaction pathway.

In the present study, the production of propylene through oxidative dehydrogenation of propane with CO_2_ was investigated over composite metal oxides M_x_O_y_-TiO_2_ (M: Ce, Zr, Ca, Cr, Ga). The influence of the nature of the M_x_O_y_ additive on the physicochemical properties of TiO_2_ was also explored, employing detailed characterization of the catalysts. An attempt was made to correlate these properties with catalytic performance in order to develop active and selective catalysts towards propylene production. DRIFTS studies were also carried out aiming to identify the surface intermediate species formed under reaction conditions and determine the beneficial effect of M_x_O_y_ modifier on reactants’ activation.

## 2. Materials and Methods

### 2.1. Catalysts Synthesis and Characterization

The incipient wetness impregnation method was used to synthesize the composite metal oxides 10% M_x_O_y_-TiO_2_. Commercial TiO_2_ (Evonik, Industries AG, Essen, Germany) was used as support, whereas the precursor salts of M_x_O_y_ were Ce(NO_3_)_3_·6H_2_O (Alfa Aesar, Kandel, Germany), ZrO(NO_3_)_2_·6H_2_O (Sigma-Aldrich, Darmstadt, Germany), Ca(NO_3_)_2_·4H_2_O (Thermo Scientific, Waltham, MA, USA), Ga(NO_3_)_3_·6H_2_O (Sigma Aldrich, Darmstadt, Germany) and Cr(NO_3_)_3_ (Thermo Scientific, Waltham, MA, USA). After impregnation, the samples were dried at 120 °C overnight and subsequently calcined in air at 600 °C for 3 h.

Nitrogen adsorption at 77 K (B.E.T. method) was applied to measure the specific surface area (SSA) of composite metal oxides using a Gemini III 2375 instrument (Micromeritics, Norcross, GA, USA). The X-ray diffraction (XRD) patterns of M_x_O_y_-TiO_2_ samples were carried out on a Bruker D8 Advance instrument (Billerica, MA, USA) operating with Cu *K_a_* radiation (*λ* = 0.15496 nm, 40 kV, 40 mA). All samples were scanned from 20 to 80° with a scan rate of 0.05°/s and a step size of 0.015°. The diffraction peaks were identified by comparing them with those provided by the JCPDS database. Scherrer’s Equation (12) was used to estimate the mean crystallite size of TiO_2_ (*d*_TiO2_):(12)$$d_{M_{x}O_{y}}=\frac{0.9\cdot \lambda }{Β\cdot cos\theta }$$
where *λ* = 0.15406 nm is the X-ray wavelength corresponding to Cu*K_a_* radiation, *B* is the peak width at half maximum intensity (in radians) and *θ* is the diffraction angle corresponding to the peak broadening. The anatase content (*x_A_*) of TiO_2_ was estimated using the following equation [24]:(13)$$x_{A}=\frac{1}{1+1.26\times (\frac{I_{R}}{I_{A}})}$$
where *I*_A_ and *I*_R_ denote the integral intensities of the peaks corresponding to (1 0 1) and (1 1 0) Miller reflections of anatase and rutile, respectively.

The basicity of metal oxides was investigated by temperature programmed desorption of CO_2_ (CO_2_-TPD) using an apparatus which consisted of a flow measuring and control system, and an electrical furnace where a fixed bed quartz reactor was placed with its outlet being directly connected to an Omnistar (Pfeiffer Vacuum, Asslar, Germany) mass spectrometer (MS). The experimental procedure involved the heating of 150 mg of catalyst at 450 °C in He where it remained for 15 min. The temperature was then decreased to 25 °C and the flow was switched to 1% CO_2_/He mixture for 30 min. A 30 min purging period with He was then followed before temperature was increased up to 750 °C using a linear heating rate of 10 °C/min.

Similar experiments were carried out employing in situ diffuse reflectance infrared Fourier transform spectroscopy (in situ DRIFTS). These experiments were conducted in a FTIR (Nicolet iS20, Thermo Fischer Scientific, Waltham, MA, USA) spectrometer equipped with an MCT detector, a KBr beam splitter and a diffuse reflectance cell (Specac, Orpington, UK). An apparatus consisting of mass flow controllers and a set of valves was directly connected into the inlet of the DRIFT cell. In these experiments, the catalyst powder was placed in the DRIFT cell and heated at 450 °C under He flow for 60 min. The temperature was then decreased to 25 °C under the same atmosphere and the catalyst was exposed to 5% CO_2_ (in He) for 30 min followed by purging with He for 10 min. The first FTIR spectrum was then collected and the temperature was subsequently stepwise increased to 450 °C. The catalyst remained at each temperature for 3 min prior to spectrum recording. All spectra were normalized by subtracting background spectra recorded in the He flow at the corresponding temperature during cooling of the catalyst. A total flow rate of 30 cm^3^/min was used in all stages of the experiment.

Temperature programmed reduction with hydrogen (H_2_-TPR) was performed to investigate the reducibility of the synthesized composite metal oxides using the apparatus described above for the CO_2_-TPD experiments. An amount of 200 mg of catalyst was loaded in a fixed bed quartz reactor and heated at 450 °C in He flow for 15 min followed by treatment at 300 °C using a mixture consisting of 20.5% O_2_/He. After being maintained under these conditions for 30 min, the temperature was increased to 450 °C in He flow and then decreased to 25 °C. A mixture of 3% H_2_/He was then introduced into the reactor and a heating program was initiated (after remaining at 25 °C for 15 min), increasing up to 750 °C using a temperature rising rate of 10 °C/min. The transient-MS signal at *m*/*z* = 2 (H_2_) was continuously monitored by the aforementioned mass spectrometer. The total flow rate in all stages was 30 cm^3^/min.

The acidity of the synthesized catalysts was investigated by conducting thermogravimetric analysis (TGA) experiments which were carried out using a TA Q50 thermal analysis instrument (TA instruments/WATERS, New Castle, Delaware). An amount of 40 mg of dried catalyst was suspended in 10% NH_3_/H_2_O solution (Merck KGaA, Darmstadt, Germany) for 1 h at 25 °C until saturation, followed by filtration and drying under vacuum at 60 °C for 1 h in order to remove water and weakly adsorbed ammonia. Thermogravimetric analysis (TGA) was then initiated by increasing temperature from 25 to 600 °C under N_2_ atmosphere using a heating rate of 10 °C/min.

### 2.2. In Situ FTIR Spectroscopy under Reaction Conditions

In situ DRIFTS studies were also performed under conditions of oxidative dehydrogenation of propane with CO_2_ using the experimental setup described above. In these experiments, the following procedure was employed: heating in He flow at 500 °C for 30 min → cooling at 25 °C in He flow → switching of the flow to 1% C_3_H_8_ + 5% CO_2_ (in He) → stepwise increasing of temperature up to 500 °C. An equilibration time of 15 min at each temperature took place prior to spectrum collection.

### 2.3. Catalytic Performance Experiments

Catalytic performance tests were carried out in a tubular fixed-bed quartz reactor (O.D.: 6mm) in the temperature range of 570–750 °C and atmospheric pressure. The catalyst sample (particle diameter: 0.15 < d_p_ < 0.25 mm) was placed in an expanded section of 5 cm length (O.D.: 12 mm) in the middle of the reactor, whereas a K-type thermocouple running through the reactor served for measuring the temperature of the catalyst bed. The reactor was placed in an electric furnace with its inlet being connected with a flow measuring and control system. The feed gases were provided by high-pressure gas cylinders and controlled by mass flow controllers. The outlet of the reactor was directly connected with a gas chromatograph (Shimadzu 2014, Kyoto, Japan) equipped with TCD and FID detectors and two packed columns (Porapak-Q, Carboxen) for the analysis of the effluent gases. A carboxen column was used for separation of CO, CO_2_ and CH_4_ in the TCD detector, whereas a Porapak-Q column was used for the separation of C_3_H_8_, C_3_H_6_, C_2_H_6_ and C_2_H_4_ in the FID detector. In these experiments, 0.5 g of catalyst was introduced to the reactor and heated under He flow at 450 °C where it remained for 1 h. The catalyst was then exposed to the feed gas mixture consisting of 5% C_3_H_8_ + 25% CO_2_/He using a total flow rate of 50 cm^3^/min, and the concentration of gas products and unreacted C_3_H_8_ and CO_2_ were analyzed by the gas chromatograph described above. Similar measurements were obtained at selected temperatures up to 750 °C.

The propane conversion ($X_{C_{3}H_{8}}$), product selectivity (*S_Cn_*), and propylene yield ($Y_{C_{3}H_{6}}$) were calculated according to the following equations:(14)$$X_{C_{3}H_{8}}=\frac{{\left[C_{3}H_{8}\right]}_{in}\cdot F_{in}-{\left[C_{3}H_{8}\right]}_{out}\cdot F_{out}}{{\left[C_{3}H_{8}\right]}_{in}\cdot F_{in}}\text{×}100$$
(15)$$S_{C_{n}}=\frac{[C_{n}]\cdot n}{\left[CO\right]+\left[{CH}_{4}\right]+2\cdot \left(\left[C_{2}H_{4}\right]+\left[C_{2}H_{6}\right]\right)+3\cdot (\left[C_{3}H_{6}\right]}\text{×}100$$
(16)$$Y_{C_{3}H_{6}}={(X}_{C_{3}H_{8}}\cdot S_{C_{3}H_{6}})/100$$
where [C_3_H_8_]*_in_* and [C_3_H_8_]*_out_* are the *v/v* concentrations (molar fractions) of C_3_H_8_ in the inlet and outlet of the reactor, respectively; *F_in_* and *F_out_* are the total flow rates (mol/s) in the inlet and outlet of the reactor, respectively; *n* is the number of carbon atoms in the corresponding molecule (e.g., 1 for CO, 2 for C_2_H_4_, 3 for C_3_H_6_, etc.) and [CO], [CH_4_], [C_2_H_4_], [C_2_H_6_] and [C_3_H_6_] are the *v/v* concentrations of the produced CO_,_ CH_4,_ C_2_H_4_, C_2_H_6_ and C_3_H_6_, respectively.

## 3. Results and Discussion

### 3.1. Catalyst Characterization

The SSAs measured following the Brunauer–Emmett–Teller (BET) method for the synthesized 10% M_x_O_y_-TiO_2_ and bare TiO_2_ samples are presented in Table 1. It was observed that the addition of a metal oxide on the surface of TiO_2_ generally resulted in a slight decrease in the SSA from 36.9 m^2^/g (bare TiO_2_) to 33.8 m^2^/g (CeO_2_-TiO_2_), with the exception of ZrO_2_-TiO_2_ and Ga_2_O_3_-TiO_2_ catalysts which exhibited an increase in the SSA of up to 47.9 m^2^/g. The decrease in the SSA was most possibly due to the partial blockage of the titania pores induced by the presence of M_x_O_y_ on its surface, in agreement with previous studies over composite metal oxides [25,26,27,28]. On the other hand, the observed increase in the SSA for the ZrO_2_-TiO_2_ and Ga_2_O_3_-TiO_2_ catalysts was previously reported to be related to the additional porosity of the particles of the metal oxide additive which may have smaller interaction with the support, as suggested by Daresibi et al. [29] and Shimizu et al. [30] over alumina-supported gallium oxide catalysts.

The X-ray diffractograms obtained for the TiO_2_-based oxides are presented in Figure 1. In the case of bare TiO_2_, the XRD pattern (trace a) consisted of peaks located at 2θ equal to 25.36°, 36.95°, 37.81°, 38.51°, 48.09°, 53.93°, 55.14°, 62.75°, 70.36°, 75.15° and 76.16°, attributed to (1 0 1), (1 0 3), (0 0 4), (1 1 2), (2 0 0), (1 0 5), (2 1 1), (2 0 4), (2 2 0), (2 1 5) and (3 0 1) indices, respectively, of tetragonal anatase (JCPDS Card No. 4-477). Crystallographic peaks at 27.42°, 36.09°, 39.21°, 41.28°, 44.15°, 54.38°, 56.70°, 62.80°, 64.11°, 69.0° and 69.86° diffraction angles corresponding to (1 1 0), (1 0 1), (2 0 0), (1 1 1), (2 1 0) (2 1 1), (2 2 0), (0 0 2), (3 1 0), (3 0 1) and (1 1 2) planes, respectively, of tetragonal rutile (JCPDS Card No. 21-1276) were also recorded. The same peaks were also detected for all the investigated composite M_x_O_y_-TiO_2_ samples (traces b–f) indicating that both anatase and rutile phases still coexisted upon the addition of metal oxide particles on the TiO_2_ surface (Figure 1). It should be noted that the intensity of certain peaks was too low for some composite metal oxides, that were hardly discernable in the obtained diffractograms. Furthermore, the XRD pattern of CaO-TiO_2_ (trace d) was also characterized by peaks at 2θ equal to 33.18°, 37.09°, 40.99°, 47.60°, 59.02° and 59.37° corresponding to (1 2 1), (2 0 2), (0 0 2), (0 4 0), (2 4 0) and (0 4 2) facets of orthorhombic CaTiO_3_ perovskite structure (JCDPS Card No. 22-153). Two crystallographic peaks at 28.73° and 33.32° attributed to (1 1 1) and (2 0 0) planes, respectively, of cubic CeO_2_ (JCPDS Card No. 1-800) were also identified for the CeO_2_-TiO_2_ catalyst (trace e). For the rest of the catalysts investigated, no other diffraction peaks were observed apart from those of the titania support, implying that Ga_2_O_3_, Cr_2_O_3_ and ZrO_2_ were well dispersed.

Anatase and rutile crystallite sizes were estimated employing the Scherrer Equation (12) using data from the peaks located at 25.36° and 27.42° diffraction angles, respectively, and the estimated values are listed in Table 1. It was observed that the mean crystallite size of the rutile phase ($d_{{TiO}_{2},R}$) decreased from 36.8 to 14.9 nm in the order TiO_2_ (bare) < CeO_2_-TiO_2_ < CaO-TiO_2_ < ZrO_2_-TiO_2_ < Cr_2_O_3_-TiO_2_ < Ga_2_O_3_-TiO_2_. A smaller decrease from 22.5 (bare TiO_2_) to 18.3 nm (Ga_2_O_3_-TiO_2_) following the same order was found for the mean anatase crystallite size ($d_{{TiO}_{2},A}$), with the exception of the Cr_2_O_3_-TiO_2_ sample where a similar crystallite size (22.7 nm) to that of TiO_2_ (bare) and CeO_2_-TiO_2_ samples was estimated. The addition of metal oxides on the TiO_2_ surface also influenced the anatase content, x_A_, calculated via Equation (13), which ranged between 59% (for bare TiO_2_) and 83% (for ZrO_2_-TiO_2_ and Cr_2_O_3_-TiO_2_).

### 3.2. Adsorption/Desorption Characteristics of CO_2_ by In Situ DRIFTS

The adsorption/desorption characteristics of CO_2_ were investigated by conducting in situ DRIFTS experiments, and results obtained over TiO_2_ and M_x_O_y_-TiO_2_ catalysts are shown in Figure 2. In the case of the bare TiO_2_ catalyst (Figure 2a), the spectrum recorded at 25 °C (trace a) in He flow following catalyst interaction with 5% CO_2_/He for 30 min was characterized by several bands in the region of 1700–1100 cm^−1^ due to (bi-)carbonates or carboxylate species adsorbed on the TiO_2_ surface. Specifically, the bands located at 1668 and 1247 cm^−1^ were previously attributed to the asymmetrical (*ν*_as_(O-C-O)) and symmetrical (*ν*_s_(O-C-O)) stretching modes of the carboxylate (CO^2−^) species, respectively [31,32,33,34], whereas the shoulder detected at 1635 cm^−1^ could be assigned to the *ν*_as_(O-C-O) mode of adsorbed bicarbonate (HCO_3_^−^) species on the TiO_2_ surface [31,32,33,34,35,36,37]. The bands located at 1405 and 1222 cm^−1^ attributed to the *ν*_s_(O-C-O) and *δ*(C-OH) modes, respectively, of bicarbonate species [31,32,33,34,35,36,38,39], were found to be formed via interaction of CO_2_ with the basic surface hydroxyl groups of TiO_2_ support [31,32,38,40]. The involvement of surface hydroxyl groups on bicarbonates formation was confirmed by the appearance of three negative bands at 3718, 3674, and 3611 cm^−1^ (not shown here) previously assigned to consumption of surface hydroxyl groups [41,42]. The bands detected at 1572 and 1351 cm^−1^ could be attributed to the bidentate form of adsorbed carbonate (CO_3_^2−^) species on the TiO_2_ surface [31,32,33,34,35,37,38,39]. Liu et al. [35] reported that bidentate carbonates were formed on a TiO_2_ surface through CO_2_ interaction with acid–base pair sites (cus Ti^4+^–O^2–^ centers).

An increase in temperature to 100 °C (trace b) resulted in a significant decrease in the intensities of the bands at 1668, 1572, 1405 and 1247 cm^−1^, while the band at 1222 cm^−1^ disappeared probably due to desorption of the corresponding species from the catalyst surface. In addition, two new bands appeared at 1522 and 1447 cm^−1^ due to monodentate carbonate [31,34,35,37,39] and bicarbonate species [32,36,38,39], respectively. The latter bands may also have been present in the spectrum recorded at 25 °C but not able to be distinguished due to overlapping, or may have developed as a result of conversion of carboxylates and/or bidentate carbonates.

Further increase in temperature to 200 °C (trace d) resulted in a progressive decrease in the intensity of all bands apart from that located at 1635 cm^−1^ which increased and maximized at 200 °C, most possibly due to conversion of the aforementioned species to bicarbonates. All bands disappeared from the spectra detected above 350 °C indicating their complete desorption from the TiO_2_ surface.

Similar experiments were conducted over M_x_O_y_-TiO_2_ (M: Ga, Cr, Zr, Ce, Ca) catalysts, and results obtained are presented in Figure 2b–f. As can be seen, adsorbed surface species over the ZrO_2_-TiO_2_ (Figure 2b) sample seemed to be similar to those discussed above over bare TiO_2_ catalyst, with the main differences being related to the variation of the relative intensities of the corresponding bands. Specifically, the spectrum recorded at 25 °C (trace a) was characterized by bands attributed to carboxylate (1670 and 1246 cm^−1^), bicarbonate (1653, 1417 and 1223 cm^−1^), bidentate carbonate (1578 and 1339 cm^−1^) and monodentate carbonate (1517 cm^−1^) species adsorbed on TiO_2_. It should be noted that bidentate and monodentate carbonates as well as bicarbonates and bridged polydentate carbonate species adsorbed on a ZrO_2_ surface give rise to the development of bands at similar wavenumbers, indicating that these species may partially contribute to the bands detected at 1655, 1578, 1517, 1417, 1339 and 1222 cm^−1^ [43,44,45]. Interestingly, the relative intensity of the bands due to bicarbonates and monodentate carbonates over 10% ZrO_2_-TiO_2_ catalyst was higher compared with bare TiO_2_, which may be related to the creation of more basic sites upon ZrO_2_ addition on the TiO_2_ surface [40]. However, adsorbed surface species on 10% ZrO_2_-TiO_2_ catalyst seemed to desorb at similar temperatures with bare TiO_2_.

In the case of the Cr_2_O_3_-TiO_2_ (Figure 2c) catalyst, bicarbonate (1633 and 1416 cm^−1^) and bidentate (1556 and 1391 cm^−1^) carbonate species associated with TiO_2_ were detected on the catalyst surface at 25 °C (trace a). Specific bands (1633, 1591, 1556, 1416 and 1358 cm^−1^) detected in the spectra obtained across the entire temperature range may also be related to species associated with Cr_2_O_3_ [46,47]. For example, Zecchina et al. [46] conducted CO_2_ adsorption experiments over a-Cr_2_O_3_ and attributed similar bands detected at 1620 and 1425 cm^−1^ to bicarbonate species, and bands at 1635, 1590, 1560 and 1340 cm^−1^ to bidentate carbonate species adsorbed on Cr_2_O_3_. Regarding desorption of surface species from the Cr_2_O_3_-TiO_2_ surface (Figure 2c), it seems that it was completed at higher temperatures (~400 °C) compared with TiO_2_ (Figure 2a) and ZrO_2_-TiO_2_ (Figure 2b) catalysts.

DRIFT spectra obtained over the CeO_2_-TiO_2_ catalyst (Figure 2d) indicated the formation of similar adsorbed surface species as those discussed above, over bare TiO_2_. However, some of the detected bands (1673, 1252, 1570–1559 and 1339 cm^−1^) may also have been associated with the formation of bidentate carbonates on the CeO_2_ surface [48,49,50,51]. It was also demonstrated that bicarbonates on CeO_2_ may have accounted for the development of bands at 1641, 1570, 1434 and 1405 cm^−1^ [48,49,50], whereas bands at 1520–1532 and 1361 cm^−1^ may have been responsible for the formation of monodentate carbonates on the CeO_2_ surface [49,50,52]. Desorption of surface species seems to occur above 400–450 °C over the CeO_2_-TiO_2_ catalyst (Figure 2d).

The main adsorbed surface species detected at 25 °C (trace a) on the surface of Ga_2_O_3_-TiO_2_ (Figure 2e) were bicarbonates (1649 and 1416 cm^−1^) and bidentate carbonates (1580 and 1320 cm^−1^) associated with a TiO_2_ [31,32,33,34,35,36,37] and/or Ga_2_O_3_ [49,53,54,55] surface. Although the same surface species were detected over Cr_2_O_3_-TiO_2_ catalysts (Figure 2c), their relative population was significantly higher, indicating that CO_2_ adsorption was favored over Ga_2_O_3_-TiO_2_. No bands due to carboxylates could be discerned. It was possible, however, that the high frequency band (~1670 cm^−1^) of carboxylate species may have been overlapped by the broad band at 1649 cm^−1^. This was also the case for the band at 1538 cm^−1^ assigned to monodentate carbonate species on TiO_2_ which was clearly discerned at 150 °C but probably pre-existed in the lower temperature spectra. The intensity of all bands decreased with increasing temperature. However, bands due to bicarbonates and bidentate carbonates were still present in the spectrum obtained at temperatures as high as 450 °C (trace i), implying that the adsorption strength of CO_2_ on the Ga_2_O_3_-TiO_2_ catalyst is high.

The population of adsorbed surface species was found to be higher over the CaO-TiO_2_ sample (Figure 2f). Carboxylates (1676 and 1241 cm^−1^), bicarbonates (1642 and 1213 cm^−1^), bidentate (1560 and 1335 cm^−1^) and monodentate (1388 cm^−1^) carbonates adsorbed on TiO_2_ were detected on the catalyst surface following CO_2_ adsorption at 25 °C. Partial adsorption of bicarbonates and monodentate carbonates on CaO may have occurred in agreement with previous studies [56,57,58]. A weak band discerned at 1769 cm^−1^ (Figure 2f) was attributed to the C=O stretching vibrational mode of a bridged-bonded carbonate species adsorbed on the CaO surface [56]. A band located at 1515 cm^−1^ was discerned in the spectrum collected at 200 °C and was assigned to an asymmetrical mode of monodentate carbonates associated with a TiO_2_ and/or CaO surface [56,59,60]. The results of Figure 2f show that a significant part of adsorbed surface species remained on the catalyst surface up to 450 °C, implying that they were strongly adsorbed on the surface of CaO-TiO_2_.

Comparison between the DRIFT spectra (Figure 2) of the investigated catalysts shows that both the population and desorption temperature of the surface species formed via interaction of catalyst with CO_2_ increased following the sequence TiO_2_ (bare) ~ ZrO_2_ < Cr_2_O_3_ < CeO_2_ < Ga_2_O_3_ < CaO. Taking into account the acidic character of CO_2_, it is expected to be preferentially adsorbed on the basic sites of metal oxides [32]. Therefore, the basicity of the investigated catalysts seems to follow the above ranking.

### 3.3. Temperature-Programmed Desorption of CO_2_

The results of the CO_2_-TPD experiments obtained over bare TiO_2_ and M_x_O_y_-TiO_2_ catalysts are presented in Figure 3. As can be seen, CO_2_ was desorbed from bare TiO_2_ exhibiting a low temperature (LT) peak centered at 72 °C, which was previously attributed to weak basic sites, and a weak high temperature (HT) peak at ca. 510 °C due to CO_2_ desorption from strong [61,62] and/or medium [63,64] basic sites.

The addition of metal oxides on the TiO_2_ surface resulted in a significant increase in the intensity of the LT peak accompanied by a shift of its maximum (by ~10 °C) towards higher temperatures following the order TiO_2_ (bare) < Cr_2_O_3_ < ZrO_2_ < CeO_2_ < Ga_2_O_3_ < CaO. Moreover, the amount of desorbed CO_2_ estimated by integrating the area below the LT peak was found to increase from 4.1 μmol g^−1^ for TiO_2_ to 32.6 μmol g^−1^ for CaO-TiO_2_ (Table 2), providing evidence that the number and strength of weak basic sites increases, following the aforementioned order. The HT desorption peak was clearly distinct for all composite metal oxides where in certain cases (Cr_2_O_3_, CeO_2_ and Ga_2_O_3_) two peaks were evolved in the high temperature range (500–750 °C). Results indicated that the number of medium and/or strong basic sites was remarkably higher in the presence of M_x_O_y_ on the TiO_2_ surface. The amount of CO_2_ desorbed at high temperatures was found to increase significantly from 0.23 μmol/g for TiO_2_ to 34.1 μmol/g for CaO-TiO_2_ (Table 2). The high strength of basic sites of CaO towards CO_2_ was also demonstrated by Constantinou et al. [60] who reported that CO_2_ was desorbed from a CaO surface above 700 °C during CO_2_-TPD experiments. In the case of the CaO-containing sample, an additional broad CO_2_ desorption peak was observed at medium temperatures (~360 °C) (Figure 3), possibly corresponding to medium basic sites.

The total amount of desorbed CO_2_ was calculated by integrating the total area below the CO_2_ response curve and was found to vary in the range of 4.3–66.7 μmol/g following the sequence TiO_2_ (bare) < ZrO_2_ < Cr_2_O_3_ ~ CeO_2_ < Ga_2_O_3_ < CaO (Table 2). As can be seen in Table 2, a similar trend was observed by comparing the amount of CO_2_ evolved per unit specific surface area (in μmol/m^2^), indicating that the observed variations in CO_2_ evolution with respect to the nature of the M_x_O_y_ additive was not a matter of SSA variation. The results of Figure 3 are in excellent agreement with the results of the DRIFTS studies discussed above (Figure 2), clearly implying that the basicity of the composite metal oxides was significantly higher than that of bare TiO_2_ and varied in a manner which depended strongly on the nature of the M_x_O_y_ additive.

An improvement of surface basicity was previously reported over titania-supported metal oxides. In particular, a shift of the CO_2_ desorption peak maximum towards higher temperatures and an increase in the amount of CO_2_ desorbed were observed by Xu et al. [15] with the addition of Ga_2_O_3_ on TiO_2_. An increase in the total basicity was also found to occur by modifying TiO_2_ with CeO_2_ [65]. Similarly, Makdee et al. [66] reported that addition of Zr improved the basicity of the Ni/TiO_2_ catalyst, favoring CO_2_ adsorption.

Regarding the strength of basic sites, Al-Shafei et al. [63] studied the basicity of ZrO_2_-TiO_2_ catalysts and observed three temperature regions of CO_2_ desorption during CO_2_-TPD, corresponding to weak (50–325 °C), medium (325–725 °C) and strong basic sites (>725 °C). They suggested that CO_2_ desorption in the low temperature range corresponded to bicarbonate species and CO_2_ evolution at medium temperatures was due to desorption of bidentate carbonates, whereas CO_2_ detection above 725 °C was related to oxycarbonates. The formation of bicarbonates on weak strength basic sites via CO_2_ interaction with OH groups was also reported by Kumar et al. [67]. However, these authors suggested that high-strength basic sites favored the formation of unidentate carbonates. This was in agreement with previous studies over TiO_2_ and ZrO_2_-TiO_2_ catalysts [40,68] where it was found that the formation of monodentate carbonates following CO_2_ adsorption occurred over strong basic sites and those of bidentate carbonates over medium basic sites, whereas bicarbonates were mainly formed over weak basic sites. Based on the results of Figure 2 bicarbonates, bidentate and monodentate carbonates were detected on the surface of all catalysts investigated following their interaction with CO_2_, with the exception of Cr_2_O_3_ where only bicarbonates and bidentate carbonates were discerned. This indicates that all types of basic sites (weak, medium and strong) possibly coexist on an M_x_O_y_-TiO_2_ surface. However, as discussed above for the results of Figure 3, CO_2_ is desorbed at two main temperature regions—low and high regions. Taking into account that the identification of the CO_2_ desorption temperature range from medium and strong basic sites is unclear in the literature [61,62,63,64,69], we cannot safely assign CO_2_ desorbed above 500 °C to medium or strong basic sites. The only certainty is that the basicity of titania is improved with the addition of metal oxides on its surface.

### 3.4. Effect of the Nature of M_x_O_y_ Additive on Catalytic Performance

The results of the catalytic performance experiments obtained over 10% M_x_O_y_-TiO_2_ catalysts for the oxidative dehydrogenation of propane with CO_2_ are presented in Figure 4, where the conversion of propane (Figure 4a) and propylene yield (Figure 4b) are plotted as a function of reaction temperature. The equilibrium conversion of propane predicted by thermodynamics was also calculated using the Outokumpu HSC Chemistry^®^ program and found to be 100% in the entire temperature range (570–750 °C) investigated (Figure 4a). According to the results of Figure 4a, bare TiO_2_ was activated above 600 °C and reached $X_{C_{3}H_{8}}$ = 50% and $Y_{C_{3}H_{6}}$ = 18% at 750 °C. The addition of a metal oxide on the TiO_2_ surface resulted in all cases in a significant improvement of catalytic performance with the propane conversion curve being shifted (by ~50 °C) towards lower temperatures. The Ga_2_O_3_- and Cr_2_O_3_-containing samples were found to be the most active catalysts, exhibiting measurable propane conversions above 550 °C and reaching $X_{C_{3}H_{8}}$ equal to 80% at 745 °C. Titania-supported ZrO_2_ and CaO samples exhibited intermediate performance and were similar to each other. Although CeO_2_-TiO_2_ was found to be more active than bare TiO_2_ below 700 °C, they presented similar propane conversions at higher temperatures.

Propylene yield was found to be strongly influenced by the nature of the metal oxide modifier and increased from 5.5 to 16% at 700 °C in the order TiO_2_ (bare) < CaO-TiO_2_ ~ CeO_2_-TiO_2_ < ZrO_2_-TiO_2_ < Cr_2_O_3_-TiO_2_ ~ Ga_2_O_3_-TiO_2_. The achievement of high yields over Ga and Cr promoted catalysts was previously reported for the production of propylene via ODP reaction [9,15,26,27,28,29,70,71,72,73,74,75]. The high activity of the Ga_2_O_3_-TiO_2_ catalyst was previously attributed to the higher number of medium strong acidic sites and the strong interaction between Ga_2_O_3_ and TiO_2_ [15]. Similarly, Xia et al. [71] found that the high surface area and the large amount of tetrahedral Ga ions which were correlated with the medium-strong Lewis acid sites, were responsible for the superior activity of the Ga_2_O_3_-Al_2_O_3_ catalyst. Moreover, Daresibi et al. [29] synthesized Ga_2_O_3_-Al_2_O_3_ catalysts using the atomic layer deposition method, and found that the dispersion of Ga_2_O_3_ on Al_2_O_3_ and the interaction between them was enhanced, leading to the formation of a higher number of Ga-O-Al linkages and higher surface moderate acidity. Τhe abundance of weak acid sites induced by the synergy between Ga_2_O_3_ and Al_2_O_3_ in the spinel-type structure of Ga_2_O_3_-Al_2_O_3_ solid solutions as well as the creation of a higher population of surface Ga sites with relative weak acidity were also reported to be responsible for the high activity and stability of the Ga_2_O_3_-Al_2_O_3_ catalyst [75]. Moreover, it was found that the activation of CO_2_ during CO_2_ conversion processes requires an optimum combination of acidic and basic properties. Lavalley et al. [76] demonstrated that the higher surface basicity of α-Ga_2_O_3_ favored the activation of CO_2_ compared with γ-Ga_2_O_3_. According to the results presented in Figure 2 and Figure 3, and as will be discussed below concerning surface acidity, the population of both basic and acid sites were increased with the addition of Ga_2_O_3_ on TiO_2_. Therefore, the high activity of the Ca_2_O_3_-TiO_2_ catalyst observed in Figure 4 can be partially attributed to the interactions between Ga_2_O_3_ and TiO_2_, which may result in an optimum number of acidic/basic sites, which seems to benefit the CO_2_-assisted ODP reaction by promoting CO_2_ and C_3_H_8_ activation on the catalyst surface.

On the other hand, the high activity and selectivity of CrO_x_ based catalysts for the ODP with CO_2_ reaction was assigned to the structural and redox properties of CrO_x_ oxides, with the catalytic performance being significantly affected by the Cr^3+^/Cr^6+^ ratio and the facile switch of Cr^3+^ and Cr^6+^ at elevated temperatures [26,70,73,74]. In the case of supported CrO_x_ materials, three different types of chromium oxides were found to exist, namely, isolated Cr^6+^, polymeric Cr^6+^ and crystalline Cr_2_O_3_, of which the activity and selectivity towards propene formation depends on the nature of the support. For example, Wang et al. [74] demonstrated that although the polymeric Cr^6+^ oxides were more active than the isolated Cr^6+^ oxides for the ODP with CO_2_ reaction over CrO_x_/silicalite-1 catalysts, they were less selective. Polymeric Cr^6+^ oxides were also found to be more active when Al_2_O_3_ was used as a support, contrary to SBA-15 supported CrO_x_ catalysts where Cr^6+^ oxides exhibited higher activity than crystalline Cr_2_O_3_ [77]. Moreover, it was found that catalytic activity of chromium oxide-based catalysts for the ODP reaction increased with increasing chromium oxide dispersion. Although the oxidation state of Cr could not be revealed based on the characterization results of the present study, the absence of XRD peaks related to CrO_x_ species from Figure 1 indicated that their dispersion on TiO_2_ surface was high and maybe (at least in part) responsible for the observed superior catalytic activity (Figure 4). Comparison of the results of the present study with the literature results over composite metal oxides is shown in Appendix A. The observed differences in catalytic activity compared with the results of the present study may be attributed to the different reaction conditions used including the mass of catalyst, the total flow rate and the CO_2_:C_3_H_8_ ratio, as well as the different catalysts’ composition, precursor compounds and synthesis method.

The distribution of products results are presented in Figure 5, where it was observed that C_3_H_6_, CO, C_2_H_4_ and CH_4_ were detected for all catalysts examined, whereas in certain cases (CaO-TiO_2_, Cr_2_O_3_-TiO_2_ and Ga_2_O_3_-TiO_2_) traces of C_2_H_6_ were also produced. In the case of bare TiO_2_ (Figure 5a), selectivity towards C_3_H_6_ ($S_{C_{3}H_{6}}$) decreased from 60 to 35% with the temperature increasing from 605 to 750 °C. This was also the case for CO selectivity (*S*_CO_) which decreased from 30 to 5%. Production of both CO and C_3_H_6_ indicated that, under the present experimental conditions, the desired reaction of oxidative dehydrogenation of propane took place, whereas part of the produced CO may have been due to the RWGS reaction (3) and/or the reverse Boudouard reaction (11). Selectivities towards C_2_H_4_ ($S_{C_{2}H_{4}}$) and CH_4_ ($S_{{CH}_{4}}$) exhibited similar behavior with temperature and increased from 10 to 40% and from 0 to 20%, respectively, in the temperature range of 605–750 °C, most possibly due to enhancement of their production via C_3_H_8_ hydrogenolysis and C_3_H_8_ or C_3_H_6_ decomposition reactions (4)–(9) [4,78].

The addition of metal oxides on the TiO_2_ surface led to certain variations in the distribution of products with respect to the nature of the metal oxide additive. Titania-supported CeO_2_ and CaO catalysts exhibited significantly lower $S_{C_{3}H_{6}}$ than *S*_CO_, which remained almost stable up to 650 °C and then decreased rapidly with further increase in temperature. This implies that the RWGS and/or reverse Boudouard reactions may dominate below 650 °C over these catalysts against the ODP reaction resulting in higher production of CO in agreement with previous studies [79,80]. It should be noted that although $S_{C_{3}H_{6}}$ was lower below 650 °C than *S*_CO_ over CeO_2_-TiO_2_ and CaO-TiO_2_, it increased with increasing temperature up to 750 °C contrary to the rest of the catalysts investigated. $S_{C_{2}H_{4}}$ and $S_{{CH}_{4}}$ were also lower ($S_{C_{2}H_{4}}$ < 20%, $S_{{CH}_{4}}$ < 10%) below 650 °C over CeO_2_-TiO_2_ and CaO-TiO_2_, indicating that C_3_H_8_ hydrogenolysis and C_3_H_8_ or C_3_H_6_ decomposition were hindered. Selectivity towards C_3_H_6_ for the most active Ga_2_O_3_-TiO_2_, Cr_2_O_3_-TiO_2_ and ZrO_2_-TiO_2_ materials was found to be similar (~40% at 650 °C) to that of bare TiO_2_ indicating that despite the increase in both propane conversion and propylene yield achieved over these composite metal oxides, propylene selectivity remained almost constant.

It is of interest to note that $S_{C_{2}H_{4}}$ was always higher than $S_{{CH}_{4}}$, implying that in addition to propane cracking via reaction (7) which results in stoichiometric production of C_2_H_4_ and CH_4_, propane cracking via reaction (6) runs in parallel, producing an excess of C_2_H_4_ [3,4]. Propane decomposition through reaction (9) and/or hydrogenolysis reactions (4) and (5) as well as propylene decomposition (8) cannot be excluded. The hydrogenolysis of propane through reaction (4) was confirmed by the detection of C_2_H_6_ traces over the samples containing CaO, Cr_2_O_3_ and Ga_2_O_3_ (Figure 5c,e,f).

In an attempt to clarify the beneficial effect of the addition of metal oxides on the TiO_2_ surface, the CO_2_-assisted oxidative dehydrogenation of propane was also conducted over bare Ga_2_O_3_ and Cr_2_O_3_ catalysts. Results showed that 10% Ga_2_O_3_-TiO_2_ catalyst exhibited significantly higher $X_{C_{3}H_{8}}$ and $Y_{C_{3}H_{6}}$ compared with bare TiO_2_ and Ga_2_O_3_ (Appendix A). This was also the case for the 10% Gr_2_O_3_-TiO_2_ catalyst which was found to be more active towards propylene formation compared with bare TiO_2_ and Cr_2_O_3_ (Appendix A). Results provide evidence that a synergistic effect exists between M_x_O_y_ and TiO_2_ leading to an improvement of the catalytic activity and process efficiency concerning propylene production. This effect may be related to the basic properties of 10% Ga_2_O_3_-TiO_2_ and 10% Gr_2_O_3_-TiO_2_ which were found to be enhanced compared with bare TiO_2_ (Figure 2 and Figure 3) as well as when compared with bare Ga_2_O_3_ or Gr_2_O_3_ as evidenced by the results of the CO_2_-DRIFTS studies presented in Appendix A. As can be seen in this graph, the relative populations of bicarbonate (1633 and 1235 cm^−1^ for Cr_2_O_3_ [47,81]_,_ 1617 and 1224 for Ga_2_O_3_ [53,54,55]) and bidentate carbonate (1587 cm^−1^ for Cr_2_O_3_ [81], 1580 and 1332 cm^−1^ for Ga_2_O_3_ [53,55]) species were significantly lower for both bare Cr_2_O_3_ and Ga_2_O_3_ catalysts compared with 10% Ga_2_O_3_-TiO_2_, 10% Gr_2_O_3_-TiO_2_ and bare TiO_2_, implying weaker adsorption of CO_2_ and, therefore, lower basicity.

Comparing the results of Figure 4 with catalyst characterization results (Table 1), a correlation between the catalytic performance and the crystallite size of TiO_2_ support was found to exist. This is clearly depicted in Figure 6a where propane conversion and propylene yield measured at 700 °C are plotted as a function of the crystallite size of the rutile phase of TiO_2_. It was observed that both $X_{C_{3}H_{8}}$ and $Y_{C_{3}H_{6}}$ increased from 15 to 45% and from 6.5 to 16%, respectively, with decreasing the $d_{{TiO}_{2},R}$ from 36.8 to 14.9 nm. A similar general trend but to a lesser degree was also found between $X_{C_{3}H_{8}}$ and $Y_{C_{3}H_{6}}$, and $d_{{TiO}_{2},A}$, with the exception of the Cr_2_O_3_-containing sample which was characterized by a similar $d_{{TiO}_{2},A}$ as that estimated for bare TiO_2_. This finding indicates that propylene production via ODP with CO_2_ reaction was favored over small TiO_2_ crystallites. The rutile content was also found to be generally lower for composite metal oxides compared with bare TiO_2_ without, however, presenting any trend with respect to $X_{C_{3}H_{8}}$ and/or $Y_{C_{3}H_{6}}$.

Moreover, based on the results of the DRIFTS (Figure 2) and CO_2_-TPD (Figure 3) studies, the surface basicity was improved with the addition of metal oxides on the TiO_2_ surface, whereas according to the results of Figure 4, catalytic activity was higher over composite metal oxides characterized by moderate basicity. This is illustrated in Figure 6b where $X_{C_{3}H_{8}}$ and $Y_{C_{3}H_{6}}$ obtained at 700 °C are presented as a function of the total amount of CO_2_ desorbed during CO_2_-TPD experiments (Table 2). It was observed that both propane conversion and propylene yield increased with the increase in surface basicity, exhibiting maximum values for the Cr_2_O_3_- and Ga_2_O_3_-TiO_2_ catalysts and then decreased rapidly for the CaO-TiO_2_ catalyst which was found to contain the largest number of basic sites.

As mentioned above, in addition to surface basicity, acidity may also influence the catalytic activity for the ODP with CO_2_ reaction. According to previous studies, the total amount of the surface acid sites of TiO_2_ and especially, those characterized by medium-strong acid strength can be significantly increased with the addition of Ga_2_O_3_ [15] or ZrO_2_ [40,82], whereas low or no influence on the TiO_2_ acidity is expected by modifying TiO_2_ with CeO_2_ [83], Cr_2_O_3_ [73,84] or CaO [60,85,86]. This was further confirmed by conducting TGA experiments following ammonia adsorption at 25 °C over selected catalysts and specifically, over TiO_2_, Ga_2_O_3_-TiO_2_ and Cr_2_O_3_-TiO_2_. The results obtained are presented in Figure 7a and Appendix A where the weight loss (%) and the TGA derivative curves, respectively, are plotted as a function of temperature. In all cases, two weight loss regions were observed. The initial weight loss appearing in the temperature range of 150–250 °C could be attributed to NH_3_ desorption from weak to moderate acid sites and the weight loss initiated above 300 °C was due to NH_3_ desorption from strong acid sites [15,83,87]. A weight loss observed below 120 °C may be due to the removal of residual physisorbed water [82]. Interestingly, the weight loss was significantly higher and extended above 350 °C for the Ga_2_O_3_-TiO_2_ catalyst indicating that this sample consisted of more and stronger acid sites compared with TiO_2_ and Cr_2_O_3_-TiO_2_. This was also reflected by the acidity values (Appendix A) estimated according to Equation (S1) following the procedure described elsewhere [88], which were found to be 310.1 μmol/g for TiO_2_, 318.8 μmol/g for Cr_2_O_3_-TiO_2_ and 510.3 μmol/g for Ga_2_O_3_-TiO_2_, clearly indicating that the total surface acidity increased with the addition of Ga_2_O_3_ on TiO_2_ but was not practically affected by modifying TiO_2_ with Cr_2_O_3_. Based on the above and taking into account that both Cr_2_O_3_-TiO_2_ and Ga_2_O_3_-TiO_2_ catalysts exhibited superior activity, it can be suggested that the surface acidity may affect catalytic activity for certain catalyst formulations but is not the only parameter that determines the conversion of propane towards propylene via the ODP with CO_2_ reaction at least under the present experimental conditions.

Based on the above, it may be proposed that a balance of surface acid/base characteristics is required as was previously suggested over various CO_2_-assisted catalytic reactions [9,89]. For example, Burri et al. [89] found that the number and strength of both acidic and basic sites were higher over the TiO_2_-ZrO_2_ catalyst compared with those measured for bare TiO_2_ or ZrO_2_. According to these authors, the optimum surface acidity and basicity were responsible for the superior activity of the TiO_2_-ZrO_2_ catalyst for the oxidative dehydrogenation of ethylbenzene to styrene. In a subsequent publication, the same authors reported that the promotion of TiO_2_-ZrO_2_ with Na or K enhanced the catalyst surface basicity, with the Na-doped sample exhibiting the optimum balance of acid/base properties leading to higher catalytic activity [90]. In addition, Sui et al. [91] prepared Cr/Na-ZSM-5 catalysts of variable Cr content for the reaction of the oxidative dehydrogenation of ethane and demonstrated that the redox properties of catalysts were influenced by Cr_2_O_3_ and Na-ZSM-5 interactions, which resulted in an increase in the number of basic sites.

It is well known that catalyst reducibility is among the physicochemical properties that were found to affect ODP activity [70]. Therefore, the redox properties of selected catalysts and particularly, the least active bare TiO_2_ and the most active Ga_2_O_3_-TiO_2_ and Cr_2_O_3_-TiO_2_ samples were examined by conducting H_2_-TPR experiments. The results (Figure 7b) showed that the TPR profile of bare TiO_2_ was characterized by a peak centered at 359 °C and a weak feature extending between 515 and 635 °C, which was previously assigned to the reduction of the surface TiO_2_ [92,93,94]. The addition of Cr_2_O_3_ on TiO_2_ support resulted in the appearance of a sharp H_2_ consumption peak with its maximum located at 256 °C, which was followed by a weaker peak centered at ~343 °C. According to previous studies, the low temperature peak was due to the reduction of Cr^6+^ to Cr^5+^ and the high temperature peak to the reduction of Cr^5+^ to Cr^3+^ [95] or the reduction of chromium located deeper in the catalyst lattice [96]. On the other hand, Wang et al. [26,74] demonstrated that the low temperature peak detected in the H_2_-TPR profiles of CrO_x_/silicalite-1 and CrO_x_ dispersed on dealuminated b zeolite was due to the reduction of isolated Cr^6+^, whereas that detected at higher temperatures was due to the reduction of polymeric Cr^6+^. An additional peak was also discerned over Cr_2_O_3_-TiO_2_ (Figure 7b) at ~365 °C (overlapped with the peak centered at 343 °C) as well as a broad feature above 400 °C which, as discussed above, may be related to the reduction of surface TiO_2_. Modification of TiO_2_ support with Ga_2_O_3_ led to significant variations of the H_2_-TPR profile which consisted of two intense peaks at 338 °C and 598 °C. Similar peaks observed over Ga_2_O_3_ based catalysts were previously attributed to the reduction of well dispersed Ga species and bulk or larger Ga_2_O_3_ particles, respectively [97,98,99]. It should be noted that the peaks discussed above due to the reduction of the TiO_2_ surface may be overlapped by the high intensity peaks of Ga_2_O_3_ reduction. The total amount of consumed H_2_ was estimated from the area below the corresponding hydrogen response curves and found to be 31.1 μmol/g for TiO_2_, 51.2 μmol/g for Cr_2_O_3_-TiO_2_ and 106.5 μmol/g for Ga_2_O_3_-TiO_2_. This indicates that the reducibility was notably enhanced with the addition of Cr_2_O_3_, and especially, Ga_2_O_3_, on the TiO_2_ surface. It is worth mentioning that although these two catalysts presented similar catalytic activity (Figure 4), the reducibility of Ga_2_O_3_-TiO_2_ was twice that of Cr_2_O_3_-TiO_2_, implying that ODP activity is not solely determined by the redox properties of TiO_2_ based catalysts.

Summarizing, the synergistic effect between M_x_O_y_ and TiO_2_ seems to involve modification of the physicochemical properties of catalysts including the variation of acid/base and redox properties, as well as the anatase/rutile content and the primary crystallite size of TiO_2_ support which influence catalytic activity and propylene yield. It should be noted that the optimization of catalytic activity of the Ga_2_O_3_-TiO_2_ and Cr_2_O_3_-TiO_2_ samples, which presented superior activity, is currently under investigation by varying the CO_2_:C_3_H_8_ ratio and/or WGHSV as well as the M_x_O_y_ content in order to achieve higher propane conversions and propylene yields at temperatures of practical interest.

### 3.5. Time-On-Stream (TOS) Stability Test

The influence of reaction time on the activity, propylene yield and products selectivity for the ODP with CO_2_ reaction was investigated over the 10% Ga_2_O_3_-TiO_2_ catalyst which was among those presenting superior performance. Measurements were obtained at 710 °C and the results showed that propane conversion fluctuated between 52 and 57% during the first 25 h on stream, whereas it progressively increased up to 66.5% with further remaining of catalyst under reaction conditions up to 32 h (Figure 8a). On the other hand, propylene yield was stable for 32 h on stream ranging between 17 and 19% (Figure 8a). Products’ selectivity remained constant for 25 h on stream, taking the following values: $S_{C_{3}H_{6}}$ = 32.5–36%, *S*_CO_ = 0.5–2%, $S_{C_{2}H_{4}}$ = 40–43% $S_{{CH}_{4}}$ = 20–22.5% and $S_{C_{2}H_{6}}$~1% (Figure 8b). A slight and progressive increase in both $S_{C_{2}H_{4}}$ and $S_{{CH}_{4}}$ up to 45.5 and 24%, respectively, was observed after 32 h on stream accompanied by a parallel decrease in $S_{C_{3}H_{6}}$ to 28.5%. This implies that the undesired reactions of propane decomposition or hydrogenolysis (4)–(7) and (9) were enhanced after prolonged catalyst interaction with the reaction mixture hindering to some extent the oxidative dehydrogenation of propane (2). In general, 10% Ga_2_O_3_-TiO_2_ catalyst exhibited sufficient stability with time on stream suggesting that is a promising material for the production of propylene via CO_2_-assisted ODP reaction.

### 3.6. Oxidative Dehydrogenation of Propane with CO_2_ Studied by In Situ DRIFTS

The identification of reaction intermediates formed on the catalyst surface under reaction conditions was investigated by conducting in situ DRIFTS experiments. In the case of bare TiO_2_ (Figure 9a), the spectrum recorded at 25 °C following catalyst exposure to 1% C_3_H_8_ + 5% CO_2_/He mixture was characterized by three negative bands located at 3718, 3677 and 3611 cm^−1^ due to surface hydroxyl groups originally existing on the TiO_2_ surface, two bands at 1565 and 1357 cm^−1^ due to bidentate carbonates, two bands at 1415 cm^−1^ and 1253 cm^−1^ assigned to bicarbonates and carboxylates, respectively, and a broad band at ca. 1657 cm^−1^ containing contributions from both the latter two species [31,32,34,36,37,38]. Several bands were also discerned in the C–H stretching (*v*) region assigned to the different vibrations of propane and/or its derivatives, which can be better seen in Figure 9d. In particular, spectral features attributed to asymmetric (2980 and 2967 cm^−1^) and symmetric (2960 cm^−1^) C–H stretching vibrations in methyl groups (CH_3,ad_), as well as asymmetric (2902 cm^−1^) and symmetric (2875 cm^−1^) C–H stretching vibrations in methylene groups (CH_2,ad_) were detected [3,25,100]. Regarding the band located at 2886 cm^−1^, it was previously assigned to *ν*_s_(CH_2_)/ν_as_(CH_3_) of gaseous propane [3,100].

An increase in temperature to 100 °C resulted in better distinguishment of the two overlapping bands at ~1630–1660 cm^−1^, confirming, as suggested above, the contribution from both carboxylate (1667 cm^−1^) and bicarbonate (1639 cm^−1^) species. Further increase in temperature resulted in an increase in the relative intensity of the 1639 cm^−1^ band at the expense of the bands at 1667 and 1565 cm^−1^, implying that either an interconversion of carboxylates and bidentate carbonates towards bicarbonates occurs under reaction conditions or bicarbonates are thermally more stable. This is in agreement with results presented in Figure 2 where bicarbonates were found to have survived on the TiO_2_ surface at higher temperatures. Most of the bands detected below 1700 cm^−1^ almost disappeared from the spectra collected above 450 °C. This was also the case for the spectral features discerned in the C–H stretching region (Figure 9d). It is of interest to note that a new band appeared at 3414 cm^−1^ in the spectrum recorded at 150 °C, the intensity of which increased significantly with progressive increase in temperature to 400 °C. A similar band was previously assigned to OH surface groups raised by H_2_O adsorption which may be produced either by the main reaction of CO_2_-assisted ODP (2) and/or the RWGS reaction (3) [39,101,102].

Similar experiments were conducted for all the investigated catalysts. Representative results for the most active 10% Cr_2_O_3_-TiO_2_ and 10% Ga_2_O_3_-TiO_2_ catalysts are shown in Figure 9b and Figure 9c, respectively. The main difference observed for the Cr_2_O_3_-containing sample compared with bare TiO_2_ was that the population of adsorbed carboxylates (1676 cm^−1^), bicarbonates (1633 and 1418 cm^−1^) and bidentate carbonates (1555 and 1360 cm^−1^) was significantly higher and progressively increased with increasing temperature up to 500 °C. It is of interest to note that the relative intensity of the bands assigned to bidentate carbonates seems to be higher compared with those due to bicarbonate and carboxylate species contrary to what was observed for bare TiO_2_. This may be related to the higher anatase content observed for 10% Cr_2_O_3_-TiO_2_ (Figure 1, Table 1) and agrees well with results reported by Su et al. [32] who demonstrated that adsorbed CO_2_ on anatase phase led mainly to bidentate carbonates formation whereas on rutile phase produced mainly bicarbonates. A significantly higher intensity was also observed for the bands in the C–H stretching region, which can be clearly discerned up to 500 °C. Moreover, two new bands were detected at 1761 and 1718 cm^−1^ in the spectra obtained between 150 and 350 °C which were previously attributed to bridged carbonate species [32,103].

The relative intensity of the bands below 1700 cm^−1^ was also found to be higher over the Ga_2_O_3_-TiO_2_ catalyst (Figure 9c) as well as over CaO-TiO_2_ (Appendix A), CeO_2_-TiO_2_ (Appendix A) and ZrO_2_-TiO_2_ (Appendix A) compared with bare TiO_2_ (Figure 9a). This implies that the adsorption/activation of CO_2_ was enhanced with the addition of metal oxides on the TiO_2_ surface most possibly due to the improved basicity observed in the results of Figure 2 and Figure 3. Besides the preferential adsorption of CO_2_ on the basic sites of metal oxides, the surface basicity has been suggested to have a beneficial effect for oxidative dehydrogenation reactions because it hinders the adsorption of the produced alkenes on the catalyst surface and consequently their deep oxidation to carbon oxides or oxygenates [104]. As in the case of the Cr_2_O_3_-TiO_2_ catalyst, the relative population of bidentate carbonate species was higher than that of bicarbonates species for all composite metal oxides which as discussed above may be related to the higher content of the anatase phase. It should be also noted that CO produced via ODP reaction (Figure 5) may also be partially responsible for the formation of adsorbed carbonate-like species on the catalyst surface. The detection of bands due to CH_3,ad_ and CH_2,ad_ species at temperatures as low as 25 °C provides evidence that propane is dissociatively adsorbed on the catalyst surface where it interacts with the adsorbed CO_2_ producing the reaction products. Based on previous studies, both non-oxidative dehydrogenation and oxidative dehydrogenation may occur during catalyst interaction with the CO_2_/C_3_H_8_ mixture in a manner which depends strongly on the catalyst reducibility and/or the type of active sites [4]. In the case of catalysts containing reducible metal oxides (e.g., Cr_2_O_3_, CeO_2_, etc.) the reaction has been suggested to proceed via the one-step oxidative route (2) with CO_2_ participating in the re-oxidation of reduced metals according to the Mars–Van Krevelen mechanism. On the other hand, when irreducible metal oxides are used (e.g., Ga_2_O_3_) the reaction occurs through the non-oxidative dehydrogenation reaction (1) in combination with the RWGS reaction (3) where the role of CO_2_ is to remove the produced H_2_ and shifts the equilibrium position towards higher propylene yields.

Although results of Figure 9 contributed to the identification of the surface intemediates produced under reaction conditions and demonstrated that the activation of CO_2_ is facilitated on composite metal oxides, they cannot reveal the reaction pathway. Clearly, detailed mechanistic studies are required in order to further explore the reaction mechanism and determine the active sites and the elementary steps of the ODP with CO_2_ reaction at the M_x_O_y_-TiO_2_ surface with respect to the nature of the M_x_O_y_ additive.

## 4. Conclusions

The addition of various M_x_O_y_ (M: Zr, Ce, Ca, Cr, Ga) additives on TiO_2_ surface for the production of propylene via the ODP with CO_2_ reaction was reported herein in an attempt to determine the role of the type of M_x_O_y_ additive on both the propane conversion and propylene yield. A considerable increase in catalytic performance was found over M_x_O_y_-TiO_2_ catalysts with the $Y_{C_{3}H_{6}}$ being increased by a factor of 2.9 following the order TiO_2_ (bare) < CaO ~ CeO_2_ < ZrO_2_ < Cr_2_O_3_ ~Ga_2_O_3_. A synergistic effect between M_x_O_y_ and TiO_2_ seems to occur, resulting in modification of the surface basicity and reducibility of the investigated catalysts as well as the anatase/rutile ratio and the primary crystallite size of TiO_2_ support. Moderate surface basicity and small TiO_2_ crystallite size were found to be crucial for the efficient conversion of propane towards propylene. DRIFTS studies carried out under reaction conditions provided evidence that CO_2_ adsorption in the form of carbonate-like species is enhanced over composite metal oxides, implying that CO_2_ activation may benefit by the presence of certain metal oxide modifiers on TiO_2_ surface, leading to higher propylene yields.

## Figures and Tables

**Figure 1 nanomaterials-14-00086-f001:**
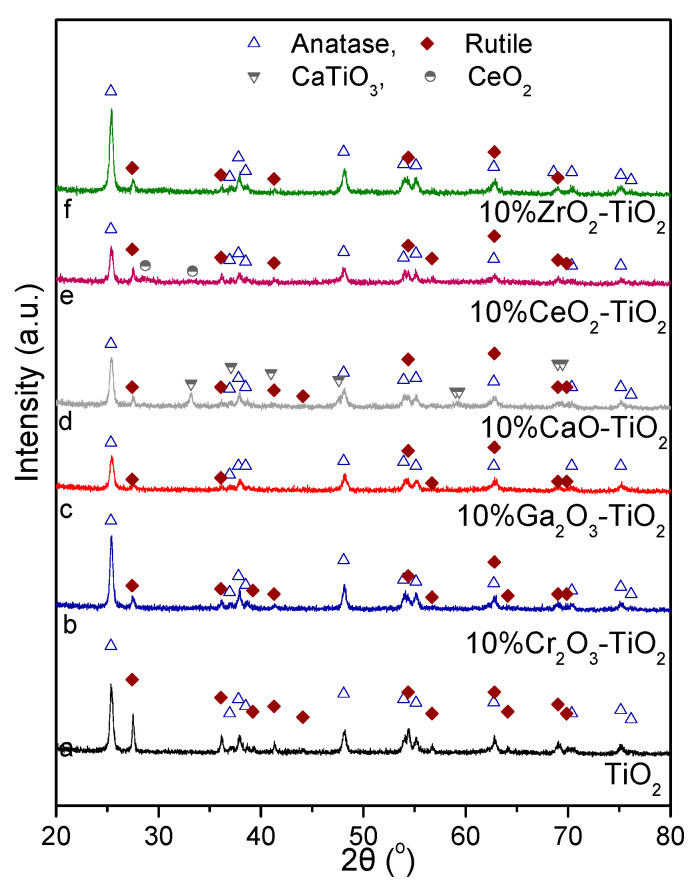
X-ray diffraction patterns obtained over TiO_2_ and 10% M_x_O_y_-TiO_2_ catalysts.

**Figure 2 nanomaterials-14-00086-f002:**
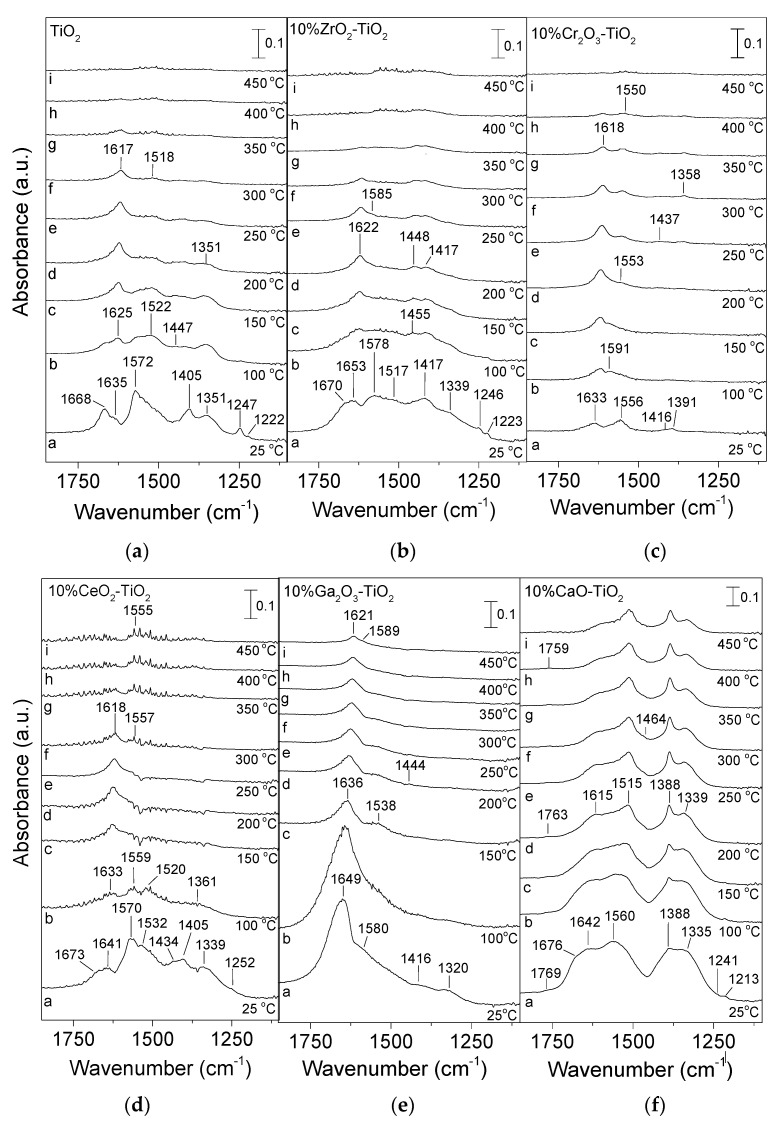
DRIFT spectra obtained from (**a**) TiO_2_, (**b**) ZrO_2_-TiO_2_, (**c**) Cr_2_O_3_-TiO_2_, (**d**) CeO_2_-TiO_2_, (**e**) Ga_2_O_3_-TiO_2_, and (**f**) CaO-TiO_2_ catalysts following adsorption of CO_2_ at 25 °C for 30 min and subsequent stepwise heating at the indicated temperatures under He flow.

**Figure 3 nanomaterials-14-00086-f003:**
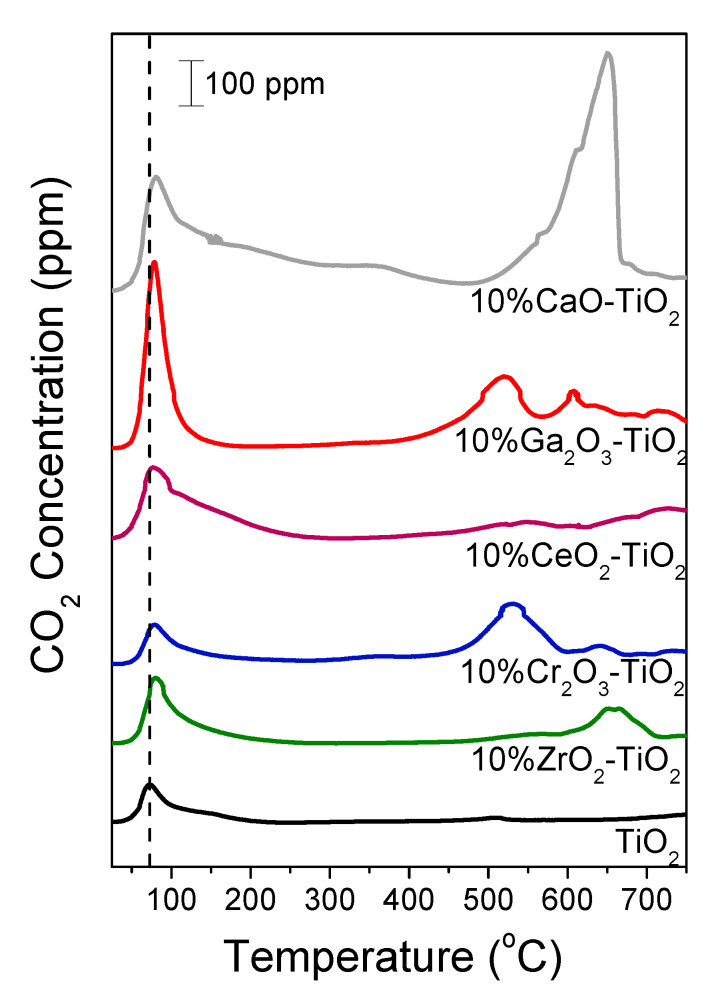
CO_2_-TPD profiles obtained from TiO_2_ and 10% M_x_O_y_-TiO_2_ catalysts.

**Figure 4 nanomaterials-14-00086-f004:**
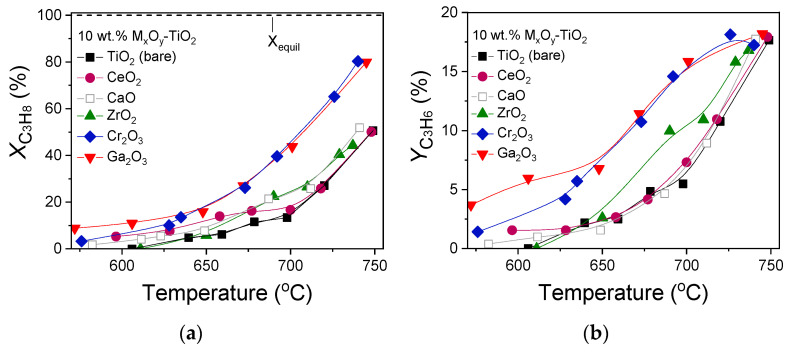
(**a**) Conversions of C_3_H_8_ and (**b**) yields of C_3_H_6_ as a function of reaction temperature obtained over TiO_2_ and 10% M_x_O_y_-TiO_2_ catalysts. Experimental conditions: mass of catalyst, 500 mg; particle diameter, 0.15 < d_p_ < 0.25 mm; feed composition, 5% C_3_H_8_, 25% CO_2_ (balance He); total flow rate, 50 cm^3^·min^−1^. Dashed line corresponds to the equilibrium conversion of propane predicted by thermodynamics.

**Figure 5 nanomaterials-14-00086-f005:**
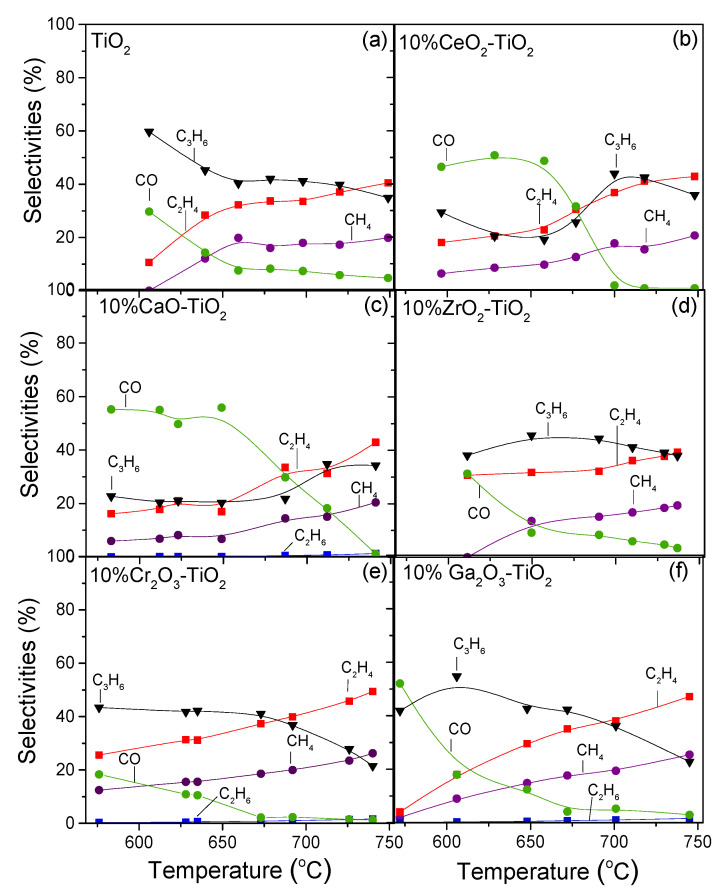
Selectivities towards reaction products as a function of reaction temperature obtained over (**a**) TiO_2_, (**b**) CeO_2_-TiO_2_, (**c**) CaO-TiO_2_, (**d**) ZrO_2_-TiO_2_, (**e**) Cr_2_O_3_-TiO_2_ and (**f**) Ga_2_O_3_-TiO_2_ catalysts. Experimental conditions: same as in Figure 4.

**Figure 6 nanomaterials-14-00086-f006:**
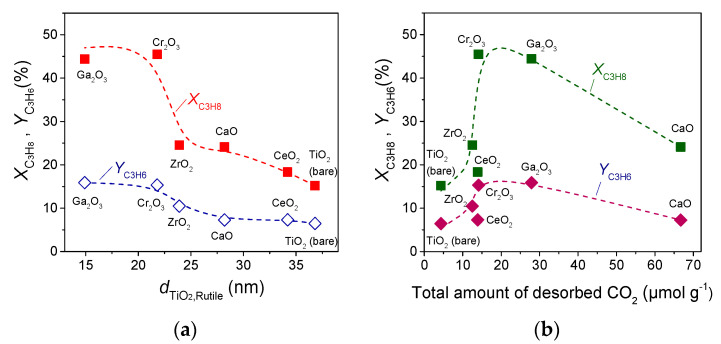
Propane conversion and propylene yield obtained at 700 °C as a function of (**a**) the mean crystallite size of rutile TiO_2_ and (**b**) the total amount of desorbed CO_2_ during CO_2_-TPD experiments of the indicated catalysts.

**Figure 7 nanomaterials-14-00086-f007:**
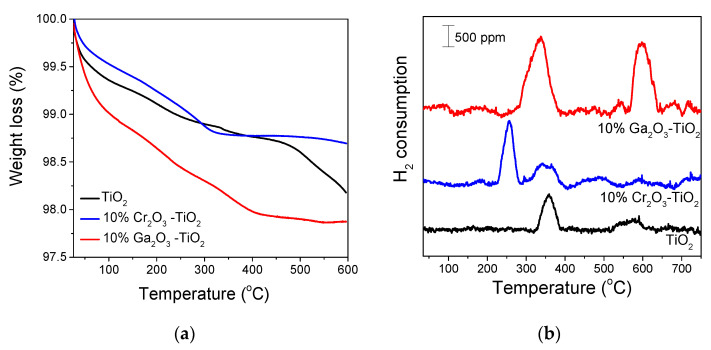
(**a**) TGA curves following NH_3_ adsorption at 25 °C and (**b**) H_2_-TPR profiles obtained from the TiO_2_, Cr_2_O_3_-TiO_2_ and Ga_2_O_3_-TiO_2_ catalysts.

**Figure 8 nanomaterials-14-00086-f008:**
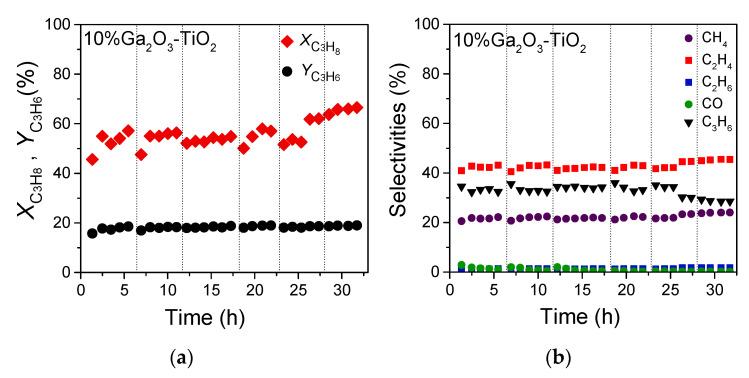
Thirty-two hour TOS stability test of the 10% Ga_2_O_3_-TiO_2_ catalyst conducted at 710 °C under conditions of oxidative dehydrogenation of C_3_H_8_ with CO_2_. Alterations of (**a**) *X*_C3H8_ and *Y*_C3H6_, and (**b**) products selectivity with time-on-stream. Experimental conditions: same as in Figure 4. Dashed vertical black lines indicate shutting down of the system overnight where the catalyst remained under He flow.

**Figure 9 nanomaterials-14-00086-f009:**
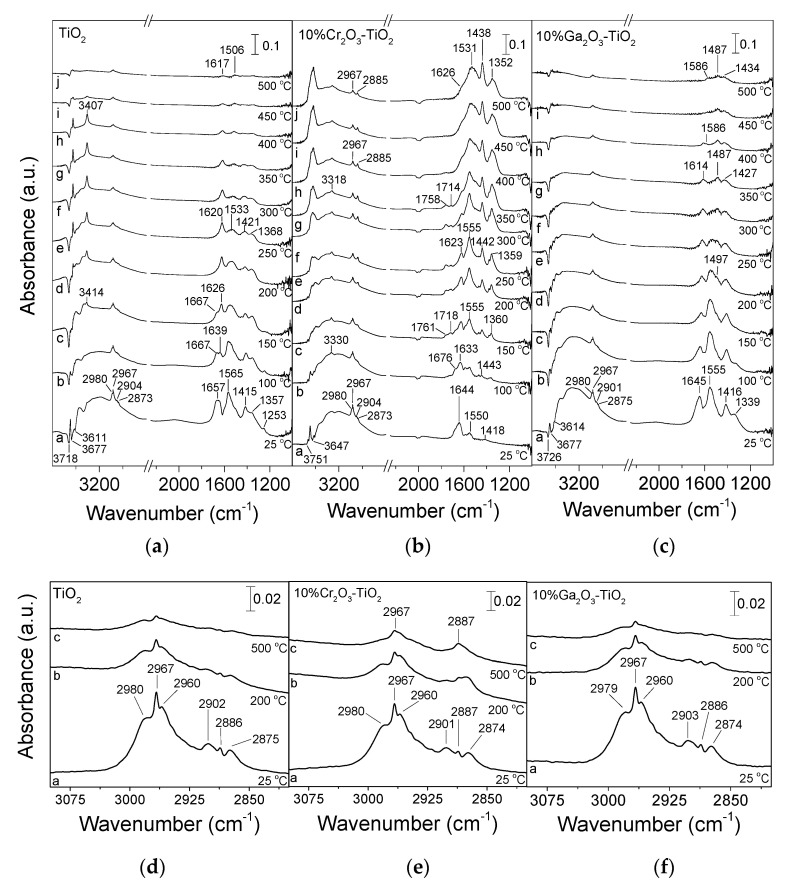
DRIFT spectra obtained over (**a**) TiO_2_, (**b**) Cr_2_O_3_-TiO_2_ and (**c**) Ga_2_O_3_-TiO_2_ catalysts following interaction with 1% C_3_H_8_ + 5% CO_2_ (in He) at 25 °C for 15 min and subsequent stepwise heating at 500 °C. The corresponding DRIFT spectra obtained in the 3100–2750 cm^−1^ region are presented in (**d**–**f**).

**Table 1 nanomaterials-14-00086-t001:** Physicochemical characteristics of the synthesized oxides.

Catalyst	SSA ^1^ (m^2^/g)	Crystallite Size ^2^ (nm)		Anatase Content ^3^ (%)
		Anatase *d*_TiO2,A_	Rutile *d*_TiO2,R_	
TiO_2_	36.9	22.5	36.8	59
10% CeO_2_-TiO_2_	33.8	22.7	34.2	66
10% CaO-TiO_2_	33.9	19.9	28.2	79
10% ZrO_2_-TiO_2_	41.7	19.9	23.9	83
10% Cr_2_O_3_-TiO_2_	36.4	22.7	21.8	83
10% Ga_2_O_3_-TiO_2_	47.9	18.3	14.9	78

^1^ Specific surface area estimated following the BET method. ^2^ Primary crystallite size of TiO_2_ estimated from the XRD line broadening. ^3^ Anatase content estimated from integral intensities of the anatase (101) and rutile (110) XRD reflections.

**Table 2 nanomaterials-14-00086-t002:** Total amount of desorbed CO_2_ during CO_2_-TPD experiments.

Catalyst	LT Peak (μmol/g)	HT Peak (μmol/g)	Total Amount of Desorbed CO_2_ (μmol/g)	Total Amount of Desorbed CO_2_ (μmol/m^2^)
TiO_2_	4.1	0.23	4.3	0.12
10% ZrO_2_-TiO_2_	8.0	4.5	12.5	0.30
10% Cr_2_O_3_-TiO_2_	4.8	9.3	14.1	0.39
10% CeO_2_-TiO_2_	12.6	1.3	13.9	0.41
10% Ga_2_O_3_-TiO_2_	13.3	14.6	27.9	0.58
10% CaO-TiO_2_	32.6	34.1	66.7	1.97

## Data Availability

Data are contained within the article and Appendix A.

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
