# Peer review of "Propylene Production via Oxidative Dehydrogenation of Propane with Carbon Dioxide over Composite MxOy-TiO2 Catalysts"

_nanomaterials, 2023, doi:10.3390/nano14010086_

Round 1
Reviewer 1 Report
Comments and Suggestions for Authors
In this manuscript the authors studied propane dehydrogenation to propylene with carbon dioxide over a series MxOy-TiO2 catalysts. They found that certain basicity of the composite metal oxides is related to their high catalytic performance. However, the reaction of propane dehydrogenation to propylene over MxOy-TiO2 catalysts is related the acidity, basicity and redox property of the catalysts, along with the synergetic interaction between MxOy and TiO2. Actually, in Abstract the authors also stated that “The addition of metal oxides on TiO2 surface results in a significant improvement of catalytic performance induced by a synergetic interaction between MxOy and TiO2 support”. However, they completely ignore the acidity, redox property, and synergetic interaction as they did not provide any experimental results, such as NH3-TPD, pyridine-IR, H2-TPR and XPS, and relevant discussion for a comprehensive study of the MxOy-TiO2 catalysts. Therefore, the current manuscript is not ready for considering further review.
In addition, the manuscript is too tedious and should be simplified significantly.
Comments on the Quality of English Languagenone
Reviewer 2 Report
Comments and Suggestions for Authors
In this work, several MxOy-TiO2 (M=Zr, Ce, Ca, Cr, Ga) catalysts were prepared for the CO2-oxidative dehydrogenation of propane (ODP). It is found that Cr2O3-TiO2 and Ga2O3-TiO2 catalysts are superior to other catalysts. Questions:
(1) The reaction temperature is too high. For the active catalysts for propane–ODP, the temperatures lower than 650C are probable. In addition, due to the high reaction temperatures, C3H8 hydrogenolysis and products decomposition took place, thus lower the selectivity of C3H6.
(2) In Figure 6, the propane conversion / products yield is correlated with the total amount basicity. It is not enough. The acidity is also important for activation of reactants.
(3) The comparison in activity of present catalysts with that reported in references is required.
Comments on the Quality of English LanguageNo.
Reviewer 3 Report
Comments and Suggestions for Authors
Reviewer 4 Report
Comments and Suggestions for Authors
This paper prepared catalysts modified with various oxides on TiO2 surface for the production of propylene by ODP reaction with CO2 and evaluated the effect of additives on propane conversion and propylene yield. At the same time, the authors were working to clarify the factors contributing to the conversion and reaction selectivity through solid base evaluation and DRIFT measurements.
It is interesting to note that the effects of additives on propane conversion and propylene yield clearly differ, with Cr or Ga being superior. The authors attribute the high activity to the moderate basicity of the TiO2 surface and small TiO2 crystallite size. However, the CO2 adsorption evaluated by CO2-TPD employed in this paper is per unit mass of catalyst. Solid basicity is mainly determined by the number of base points on the particle surface and is not easily affected by the bulk proerties. Why else did the authors use CO2 adsorption per mass as an indicator of solid basicity? Shouldn't the amount of CO2 adsorbed per unit specific surface area be used? Furthermore, if basicity per unit specific surface area is used as an indicator, is there any possibility that propane conversion and propylene selectivity can be simply organized by TiO2 crystallite diameter?
These questions are considered relevant to the present paper, and a sincere response from the authors is expected.
Round 2
Reviewer 1 Report
Comments and Suggestions for Authors
The authors have conducted some more characterizations and added some relevant discussion in the revised manuscript. The manuscript has been improved a bit.
However, in the file named Author’s Reply, the authors stated that “Regarding surface acidity, we agree with the reviewer that it plays a crucial role on catalytic activity for the CO2-assisted oxidative dehydrogenation of propane. A balance of surface acid/base characteristics is required in order to ensure the efficiency of the ODP with CO2 process. …… However, due to inaccessibility to instruments that could serve for the conduction of such experiments in our institutions as well as due to the lack of required reagents and the short available time (7 days) to complete revisions of the present manuscript, this was impossible”.
I really feel this is impossible to be considered as a proper scientific explanation. Firstly, if one accepts the characterization is crucial for the research in this manuscript, he/she should find another place to do it. The inaccessibility to instruments in he/she institution is not an acceptable excuse. Secondly, if there is difficulty to complete the revision within 7 days, one may ask to extend the revision period or resubmit it as a new manuscript. To submit a revised manuscript without some essential research work realized by the authors is not acceptable.
I, therefore, suggest the authors to finish all necessary work and submit this manuscript as a new one.
In addition, this manuscript is like parts of a degree thesis. The main content may be shortened significantly and the number of the references may be squeezed.
Round 3
Reviewer 1 Report
Comments and Suggestions for Authors
The manuscript may be considered for acceptance.
Comments on the Quality of English LanguageA thorough check is still suggested.